

# Reassessing the observational evidence for nitrogen deposition impacts in acid grassland: spatial Bayesian linear models indicate small and ambiguous effects on species richness

Oliver L. Pescott[1] and Mark Jitlal[2]

[1] UK Centre for Ecology & Hydrology, Wallingford, Oxfordshire, United Kingdom
[2] Queen Mary University of London, Wolfson Institute of Preventative Medicine, London, United Kingdom

## ABSTRACT

Nitrogen deposition (Ndep) is considered a significant threat to plant diversity in grassland ecosystems around the world. The evidence supporting this conclusion comes from both observational and experimental research, with "space-for-time" substitution surveys of pollutant gradients a significant portion of the former. However, estimates of regression coefficients for Ndep impacts on species richness, derived with a focus on causal inference, are hard to locate in the observational literature. Some influential observational studies have presented estimates from univariate models, overlooking the effects of omitted variable bias, and/or have used *P*-value-based stepwise variable selection (PSVS) to infer impacts, a strategy known to be poorly suited to the accurate estimation of regression coefficients. Broad-scale spatial autocorrelation has also generally been unaccounted for. We re-examine two UK observational datasets that have previously been used to investigate the relationship between Ndep and plant species richness in acid grasslands, a much-researched habitat in this context. One of these studies (*Stevens et al., 2004*, *Science*, 303: 1876–1879) estimated a large negative impact of Ndep on richness through the use of PSVS; the other reported smaller impacts (*Maskell et al., 2010*, *Global Change Biology*, 16: 671–679), but did not explicitly report regression coefficients or partial effects, making the actual size of the estimated Ndep impact difficult to assess. We reanalyse both datasets using a spatial Bayesian linear model estimated using integrated nested Laplace approximation (INLA). Contrary to previous results, we found similar-sized estimates of the Ndep impact on plant richness between studies, both with and without bryophytes, albeit with some disagreement over the most likely direction of this effect. Our analyses suggest that some previous estimates of Ndep impacts on richness from space-for-time substitution studies are likely to have been over-estimated, and that the evidence from observational studies could be fragile when confronted with alternative model specifications, although further work is required to investigate potentially nonlinear responses. Given the growing literature on the use of observational data to estimate the impacts of pollutants on biodiversity, we suggest that a greater focus on clearly reporting important outcomes with associated uncertainty, the use of techniques to account for spatial autocorrelation, and a clearer focus on the aims of a study, whether explanatory or predictive, are all required.

Corresponding author
Oliver L. Pescott, olipes@ceh.ac.uk

## INTRODUCTION

Nitrogen deposition (Ndep) is a significant threat to the plant diversity of various habitat types, both in north-western Europe (*UK National Ecosystem Assessment, 2011*) and around the world (*Phoenix et al., 2006*). The evidence for this position comes from a variety of sources, including correlative analyses of observational data (e.g., *Maskell et al., 2010*), typically conducted across relatively large areas as "space-for-time" gradient studies, and small scale experiments (e.g., *Van der Eerden et al., 1991*), although the latter have also frequently been pooled across larger areas through meta-analyses or other approaches to evidence synthesis (*Clark et al., 2007*; *Phoenix et al., 2012*; *Soons et al., 2017*). Reviews of Ndep impacts on plant biodiversity have typically drawn on all of this evidence (*Bobbink et al., 2010*; *UK National Ecosystem Assessment, 2011*; *Stevens et al., 2011c*; *RoTAP, 2012*; *Rowe et al., 2017*), in addition to other types of studies, such as Before-After surveys of historic plots (e.g., *Britton et al., 2009*). Observational and experimental studies are therefore both generally considered useful ways of understanding pollutant-driven biodiversity change in terrestrial ecosystems.

Different inferential approaches are often considered complementary, with large-scale, observational methods potentially allowing access to "treatment" effects across pre-existing gradients, with levels of replication that would likely be challenging to resource via an experimental route (but see *Fraser et al., 2013*). One cost of this approach is that the effect of interest is likely to be crossed in various complex ways with numerous other variables, including historic drivers for which data are likely to be inaccessible, leaving one with a large choice of covariates that could potentially be included in a model, including some which will be unknown, or suspected to be of importance but impossible to access. Furthermore, spatially autocorrelated variables that are not captured by the covariates included, or other processes causing spatial structure such as dispersal, may also need to be accounted for to ensure accurate estimation of regression coefficients (*Beale et al., 2010*; *Crase et al., 2014*). The ultimate purpose of a statistical model must also be taken into account in making analytical decisions: does one primarily wish to make predictions, or is the focus on unbiased effect estimation to develop causal understanding (*Mac Nally, 2000*; *Stephens, Buskirk & Del Rio, 2007*; *Shmueli, 2010*)? Although causal inference may imply predictive success, models that are constructed using methods that solely seek to maximise predictive accuracy will not necessarily capture causal processes accurately. Even when explicitly aiming for causal explanations through regression modelling in non-experimental situations, estimated coefficients may still only have a weak claim to be viewed as causal effects (*Gelman & Hill, 2007*; *Young, 2018*). Statistical issues then, in addition to domain-specific understanding, must also be at the forefront when attempting to make statements about cause and effect from observational data (*Rubin, 2004*; *Young, 2018*).

The work on Ndep presented here arose from a desire to use information from existing studies to inform the analysis of new data through the use of informative priors in a Bayesian framework (*Lemoine, 2019*). However, causality-focused estimates of regression coefficients from observational studies of Ndep impacts on plant richness proved hard to find in the current literature. For example, several studies have presented "headline" estimates from univariate models after multiple regression modelling (*Stevens et al., 2004*; *Maskell et al., 2010*; *Field et al., 2014*), implying regression coefficient values for Ndep that do not necessarily have any causal meaning, and/or have used *P*-value-based stepwise variable selection (PSVS) to derive final models, a strategy long known to be poorly suited to the accurate estimation of regression coefficients for inferential purposes (*Greenland & Neutra, 1980*; *Mac Nally, 2000*; *Whittingham et al., 2006*; *Faraway, 2014*; *Harrell, 2015*; *Heinze, Wallisch & Dunkler, 2018*). Re-analysis of existing studies with the primary aim of developing models that focus on causal inference should therefore be valuable in exploring the dependence of earlier conclusions on modelling choices. Fortunately, data focusing on UK acid grasslands from two such studies (*Stevens et al., 2004*; *Maskell et al., 2010*) were available for re-analysis; but note also that very similar work has been done across other habitats (e.g., *Maskell et al., 2010*; *Field et al., 2014*), and across larger areas (e.g., *Dupré et al., 2010*).

The main focus of the current work is on deriving estimates of the effect sizes of nitrogen deposition on plant species richness, rather than on their statistical significance (*Amrhein, Greenland & McShane, 2019*), whilst accounting for previously unmodelled broad-scale spatial autocorrelation. The model forms investigated were specified *a priori*; in general these were delimited by the full sets of covariates previously investigated by the original studies, given that these all have good ecological reasons for inclusion. We do not make the (unprovable) claim that our "full" models are fully correct with respect to reality, merely that the inclusion of as many plausible "pre-treatment" covariates as possible is likely to help avoid omitted variable bias, given that the ignorability assumption (i.e., all confounders are measured) is more likely to be satisfied (*Gelman & Hill, 2007*; *Young, 2018*), and that even our worst case ratio of response data points to predictors meets rules of thumb put forward by statisticians to help ensure that coefficients can be estimated accurately (*Harrell, 2015*, p. 72; *Heinze, Wallisch & Dunkler, 2018*). In addition, we also include a model for the *Stevens et al. (2004)* data that focuses on a reduced set of covariates chosen for their expected similarity to the ecological impacts of the covariates used by *Maskell et al. (2010)* for comparative purposes. Two additional models for the *Maskell et al. (2010)* data were specified *post hoc*, due to a desire to investigate differences between our results and those of the original paper.

Note that some covariates of interest (e.g., the topsoil variables pH, Al, C:N, and %N) could themselves be influenced by Ndep, but will also have independent impacts on the dependent variable (species richness). These types of partly intermediate variables are sometimes distinguished from fully post-treatment variables as "proxy variables" (*Angrist & Pischke, 2009*, pp. 64–68) whose inclusion is often better than their omission when a causal interpretation is desired. We assume here that, due to their pre-treatment importance for richness, adjusting for these covariates is more likely to result in accurate estimates of

the effect of Ndep than not adjusting for them (*Angrist & Pischke, 2009*); indeed, in some cases, their inclusion will be essential for avoiding bias (*Rosenbaum, 1984*). Ultimately, we assume that these potential post-treatment (or proxy) variables are "plausible surrogate[s] for … clearly relevant but unobserved pretreatment variable[s]" (*Rosenbaum, 1984*).

## MATERIALS AND METHODS

### Datasets

These are discussed chronologically by date of the field survey that created the original dataset, rather than the date of publication of the analysis. The data analysed by *Maskell et al. (2010)*, "MEA10" hereafter, are described in that paper, and were originally collected as a part of the 1998 UK Countryside Survey (UKCS; http://www.countrysidesurvey.org.uk). Briefly, the data analysed by MEA10 were selected from the 1998 UKCS on the basis of matches between plant communities in 2 × 2 m plots and particular National Vegetation Classification syntaxa; for acid grasslands, the chosen plots had a best fit to the acid grassland types U1-9 (*Rodwell, 1992*). For more details on the UKCS sampling strategy see *Maskell et al. (2010)* and *Smart et al. (2003a)*; we only note here that the UKCS is a stratified sample of so-called UK land classes (*Firbank et al., 2003*), with systematic random sampling within strata. We take the opportunity here to clarify some points relating to the datasets used in MEA10: the number of acid grassland plots used in analyses was 883, not 895 as reported; total nitrogen deposition from the modelled dataset of *Smith et al. (2000)* was specifically the estimated deposition over moorland, where moorland was defined according to the Land Cover Map 1990 (*Fuller et al., 1993*), and so is generally considered a better match to acid grassland than the grassland category of *Smith et al. (2000)*; and, finally, the sulphur dioxide deposition covariate was not absolute deposition for a particular year, but the difference between the modelled values over moorland for 1998 and the modelled peak in 1970 at a 5 × 5 km resolution (all LC Maskell & SM Smart, pers. comm., July 2019). This approach attempts to ensure that the covariate effectively measures the recent and substantial reduction in acidifying sulphur deposition across Britain.

Descriptions of the data collected by *Stevens et al. (2004)*, "SEA04" hereafter, can be found in that paper, and several others (e.g., *Stevens et al., 2011b*). The dataset reanalysed here was archived by *Stevens et al.* (*2011a*; Ecological Archives deposit E092-128). Briefly, SEA04 surveyed a random sample of 68 sites (after applying size and accessibility filters) from a larger database of acid grasslands collated by British national conservation agencies (note that the grasslands in this database may not be representative of the total national habitat resource). The random sample was stratified across a nitrogen deposition gradient, the gradient being again the deposition model of *Smith et al. (2000)*, as for MEA10. Within sites, five 2 × 2 m plots were recorded (i.e., $n = 68 \times 5 = 340$) within a larger 100 × 100 m area chosen to contain at least 50% of NVC type U4 (*Festuca ovina—Agrostis capillaris—Galium saxatile*; (*Rodwell, 1992*) acid grassland (*Stevens et al., 2004*). Note, however, that *Stevens et al. (2011a)* archived data for 320 of the plots from *Stevens et al. (2004)*, and it is this dataset that we reanalyse here.

## Data preparation

Within each dataset, some covariates were re-scaled to allow for more direct comparisons between regression coefficients, and to allow for their more intuitive ecological interpretation (*Gelman & Hill, 2007*). For example, altitude, scaled in metres in the original datasets, was divided by 100 to produce regression coefficients that estimated the change in richness per 100 m. This produces a range of 9.75 (0.00–9.75) for MEA10, comparable to the other covariates, and a more ecologically-interpretable regression coefficient (see Table 1 for all covariates used and their ranges). Although standardisation to unit variance is also often recommended for improving the comparability of coefficients in regression modelling (*Gelman & Hill, 2007*), it can also make direct comparisons between studies more difficult (*Baguley, 2009*), and so we focus here on models estimated using our rescaled covariates (where deemed necessary; see Table 1). MEA10 reported little difference in conclusions with respect to the analysis of vascular plant richness only or vascular plant plus bryophyte richness. SEA04 included bryophytes in their analyses. We focus primarily on vascular plant richness responses only (given that they are more likely to be accurately estimated; for example, MEA10 note that UKCS surveys only include ''a selected range of the more easily identifiable bryophytes''), but we also report results from analyses including bryophytes in Supplementary Information 2.

SEA04 considered a larger number of covariates than MEA10, therefore we present two reanalyses of the SEA04 dataset here: model 1, using a smaller set of covariates chosen to match those of MEA10 in terms of their likely ecological effects; and model 2, using a larger set of covariates, matching those considered by SEA04 as closely as possible. Note that the full set of covariates considered by SEA04 contains some that are perfect linear combinations of each other (e.g., total acid deposition is given by SEA04 as total N plus total S; likewise, total N is normally calculated as reduced N plus oxidised N), for this reason, we only consider total N deposition and total S deposition as pollutant covariates in our models (Table 1).

For SEA04 model 2, four of the available climate variables chosen to match the analysis of *Stevens et al. (2004)* had very high pairwise linear correlations (all with $r \geq 0.78$); these variables (mean annual potential evapotranspiration (PET), mean annual daily maximum and minimum temperatures, 1996–2006, all from the MARS dataset (see *Stevens et al., 2011b*); and mean annual potential evapotranspiration from *Tanguy et al. (2018)*) were combined using PCA and the first two principle components of this ordination used in their place (Table 1). PET data were unavailable for 3 sites (15 plots) in SEA04 (all on Lundy Island, England, 51°10′57.82″N, 4°40′11.46″W), and these values were imputed as the mean PET value across the remainder of the SEA04 dataset prior to PCA. Imputation of the missing values by predicting the missing values of PET using the other highly correlated climate variables changed the imputed value of the missing data (all 15 plots had the same values for the predictive climate variables), but did not result in any substantive change to the final regression coefficients of the two climate principle components in the spatial Bayesian models described in the next section, nor to those of other variables.

All other pairwise correlations in the SEA04 dataset were $\leq |0.57|$, except for total Ndep and total Sdep, which were also highly correlated ($r = 0.83$); a similar situation applied to
**Table 1 Summary information for all covariates used in our reanalyses.** All covariates with their original spatial grain sizes, original ranges, and re-scalings as used in the reanalyses of *Maskell et al. (2010)* (MEA10) and *Stevens et al. (2004)* (SEA04) presented here.

| Covariate | Supporting refs or sources | Original data grain size | Original range | Original units | Rescaled range where relevant | Rescaled units where relevant | Relevant reanalysis |
|---|---|---|---|---|---|---|---|
| Total Ndep estimated over moorland | *Maskell et al. (2010)* and *Smith et al. (2000)* | 5 × 5 km | 4.9–40.0 | kg ha$^{-1}$yr$^{-1}$ | – | – | |
| Change (1970 to 1998) from peak Sdep estimated over moorland | *Smith et al. (2000)* and L Maskell (pers. comm., 2019) | 5 × 5 km | −5.36–0.00 | Δ kg S ha$^{-1}$yr$^{-1}$ | – | – | |
| Max. altitude | *Maskell et al. (2010)* | 1 × 1 km | 0–975 | m | 0.00–9.75 | 100 m | MEA10 |
| Mean min. Jan. temp (1961–1999) | *Maskell et al. (2010)* | 5 × 5 km | −8.16–0.08 | °C | – | – | |
| Mean max. Jul. temp (1961–1999) | *Maskell et al. (2010)* | 5 × 5 km | 14.11–26.67 | °C | – | – | |
| Mean annual precipitation (1961–1999) | *Maskell et al. (2010)* | 5 × 5 km | 554.33–3305.80 | mm | 2.22–13.22 | 250 mm | |
| Change in sheep numbers | *Maskell et al. (2010)* | 2 × 2 km | −11.19–88.47 | Δ sheep per year (1969 to 2000) | −1.12–8.85 | Δ 10 sheep per year (1969 to 2000) | |
| Mean annual precipitation (1996–2006) | *Stevens et al. (2011a)* | 25 × 25 km | 604.9–1773.3 | mm | 0.42–7.09 | 250 mm | |
| Mean annual daily max. temp. (1996–2006) | *Stevens et al. (2011a)* | 25 × 25 km | 11.5–14.6 | °C | – | – | SEA04: Model 1 |
| Mean annual daily min. temp. (1996–2006) | *Stevens et al. (2011a)* | 25 × 25 km | 4.2–8.1 | °C | – | – | |
| Topsoil aluminium | *Stevens et al. (2011b)* | Empirical plot data | 11.60–1318.75 | mg kg$^{-1}$ dry soil | 0.06–6.59 | 200 mg kg$^{-1}$ dry soil | |
| Total Sdep estimated over grassland | *Stevens et al. (2004)* and *Smith et al. (2000)* | 5 × 5 km | 3.20–13.44 | kg ha$^{-1}$yr$^{-1}$ | – | – | |
| Total Ndep estimated over grassland | *Stevens et al. (2004)* and *Smith et al. (2000)* | 5 × 5 km | 7.70–40.86 | kg ha$^{-1}$yr$^{-1}$ | – | – | |
| Topsoil pH | *Stevens et al. (2004)* and *Stevens et al. (2011b)* | Empirical plot data | 3.69–5.37 | pH unit | 7.38–10.74 | 0.5 pH unit | SEA04: Models 1 & 2 |
| Max. altitude | *Stevens et al. (2004)* and *Stevens et al. (2011b)* | Empirical plot data | 15–500 | m | 0.15–5.00 | 100 m | |
| Grazing intensity | *Stevens et al. (2004)* and *Stevens et al. (2011b)* | Empirical plot data | Coded as low/medium/high | – | – | – | |

**PeerJ** ─────────────────────────────────────────────

| Covariate | Supporting refs or sources | Original data grain size | Original range | Original units | Rescaled range where relevant | Rescaled units where relevant | Relevant reanalysis |
|-----------|---------------------------|--------------------------|----------------|----------------|-------------------------------|-------------------------------|---------------------|
| Climate PC1 | See methods | See methods | −4.90–3.43 | – | – | – | – |
| Climate PC2 | See methods | See methods | −1.47–1.20 | – | – | – | – |
| C:N | Stevens et al. (2004) and Stevens et al. (2011b) | Empirical plot data | 13.34–30.58 | topsoil mass ratio | – | – | |
| Slope | Stevens et al. (2004) and Stevens et al. (2011b) | Empirical plot data | 0–60 | ° | 0–6 | 10° | SEA04: Model 2 only |
| Soil %N | Stevens et al. (2004) and Stevens et al. (2011b) | Empirical plot data | 0.12–1.57 | topsoil %N | – | – | |
| Soil moisture deficit (SMD) | Stevens et al. (2004) and Hough & Jones (1997) | 40 × 40 km | 1.66–48.94 | mm | 0.17–4.89 | 10 mm | |

the MEA10 data, where total Ndep and Sdep change had a correlation of $r = -0.70$ (all others were $\leq |0.68|$). However, in both cases these pollutant variables were retained in our models due to their intrinsic interest to our causal question. Collinearity should not bias regression coefficient estimates, but can lead to higher variance (*Harrell, 2015*; *Fox, 2016*); variance inflation factors (VIFs) calculated for both datasets using standard Poisson generalised linear mixed effects models (with a random intercept for each 1 kilometre square containing plots; see below) indicated that all VIFs were below 5.4. The square-root of this VIF is 2.3, below the range that *Fox* (*2016*, p. 343) indicates can present serious issues for the precision with which regression coefficients are estimated.

## Statistical models

As discussed by *Blangiardo & Cameletti (2015)*, models for point-referenced data (i.e., those with measurements of some outcome across a set of specific locations, where the locations are indexed by a two (or three) dimensional vector) are one type of spatial data that can be modelled within a Bayesian framework (such point-referenced data are also known as geostatistical data). Hierarchical approaches to regression modelling, where unstructured random effects are incorporated within the model, are often implemented using a Bayesian approach (*Gelman & Hill, 2007*; *Blangiardo & Cameletti, 2015*). Such models can be extended to include structured random effects that allow analysts to account for similarities based on temporal or spatial neighbourhoods. The integrated nested Laplace approximation (INLA) method of approximate Bayesian inference is particularly well-suited to this area of modelling given its speed and relative ease of implementation (*Blangiardo & Cameletti, 2015*).

We modelled both datasets using hierarchical Poisson regressions in R-INLA (*Rue, Martino & Chopin, 2009*; http://www.r-inla.org). All models included a set of covariates and an additional random spatial field, to account for broad-scale spatial autocorrelation, as independent variables. The spatial field was evaluated using the stochastic partial

differential equation (SPDE) approach developed by *Lindgren, Rue & Lindström (2011)*, and was specified as a mesh constructed using a triangulation based on the $1\times 1$ kilometre squares of the British national grid (EPSG identifier 27700). Our models also included a random effect for the 1 kilometre squares containing the plots (as per *Maskell et al., 2010*), because in both datasets there were instances of multiple plots being recorded within a single square. Therefore the random spatial field models broader scale spatial autocorrelation between squares, whilst the 1 km square random effect captures smaller scale autocorrelation between plots within squares (*Maskell et al., 2010*). For the MEA10 dataset, such models had considerably lower values of the Deviance Information Criterion (DIC) than models omitting a random effect of 1 km square ($\Delta DIC = 54.6$ or 31.8, without and with bryophyte data respectively); for SEA04 models 1 and 2 the inclusion of the 1 km square random effect was less important, but still generally improved the model (model 1: $\Delta DIC = 1.3$ or $-3.7$; model 2: $\Delta DIC = 2.6$ or 2.0; both without and with bryophyte data respectively). The 1 km square random effect was therefore included for all models. The same check was made on the inclusion of the random spatial field based on excluding this term from a model including the 1 km square random effect. The results indicated that the spatial random field was required in five out of six models (MEA10: $\Delta DIC = 7.4$ or 10.8, without and with bryophyte data respectively; for SEA04: model 1: $\Delta DIC = 7.4$ or $-1.1$; model 2: $\Delta DIC = 11.1$ or 10.6; again, both without and with bryophyte data respectively). The form of our final model can therefore be described as follows:

$$y_{ij} \sim \text{Poisson}(\lambda_{ij})$$

$$\lambda_{ij} = \exp\left(\beta_0 + \sum_{m=1}^{M} \beta_m X_{mij} + \beta_j Q_{j[i]} + \omega_{ij}\right).$$

where $y_{ij}$ is an observation of plant species richness in any plot/square combination, $i$ is a unique plot identifier, $j$ is a unique 1 km square identifier, $\beta_0$ is a global intercept, $\beta_m$ is the regression coefficient for covariate $X_m$ ($m = 1,\ldots,M$) and $\beta_j$ is the hierarchical coefficient for each 1 km square, with $\beta_j \sim N(0, \theta^2)$ and $Q_{j[i]}$ indicates the association of the $i$th observation with the $j$th square (*Gelman & Hill, 2007*). The $\omega_{ij}$ term represents an additional spatial effect assumed to be a zero mean Matern Gaussian Markov random field (GMRF) evaluated by the SPDE solution estimated over the specified triangulated mesh (Supplementary Information 1; Fig. A1.7). A detailed overview of geostatistical models estimated using SPDEs and GMRFs can be found in *Blangiardo & Cameletti (2015)*. Priors for all parameters were left at the INLA defaults (see http://www.r-inla.org).

## RESULTS

The inclusion of both the 1 km square random effect and the spatial random field improved almost all the models of the effects of N deposition on plant species richness in acid grassland explored here (as judged by DIC, see "Statistical models" above). The one model (SEA04 model 1 with bryophyte data) where $\Delta DIC$ was negative when either the 1 km square random effect or the spatial random field was dropped (indicating that the simplified model was preferred) still required a spatial random effect at some scale: dropping both

**Table 2  Estimated regression coefficients for total nitrogen deposition across models.** Exponentiated total nitrogen deposition regression coefficient medians, 2.5 and 97.5% quantiles for all models (all given to 2 decimal places). For example, for model MEA10, vascular plants only, a median value of 1.01 implies that a 1% gain of species richness per kg ha$^{-1}$ yr$^{-1}$ of total N deposition is highly compatible with the data; however, the 2.5% quantile value of 0.99 for this model also suggests that losses up to and around 1% are plausible (all values being conditional on model assumptions and data accuracy).

| Model, dependent richness variable | Spatial error structure | 2.5% quantile | Median | 97.5% quantile | Model location |
|---|---|---|---|---|---|
| MEA10, vascular plants only | Mesh + square | 0.99 | 1.01 | 1.03 | Fig. 1A |
| MEA10, vascular plants and bryophytes | Mesh + square | 0.99 | 1.00 | 1.02 | SI2: Fig. A2.1 |
| SEA04 model 1, vascular plants only | Mesh + square | 0.98 | 0.99 | 1.00 | Fig. 1B |
| SEA04 model 2, vascular plants only | Mesh + square | 0.98 | 0.99 | 1.01 | Fig. 1C |
| SEA04 model 1, vascular plants and bryophytes | Mesh + square | 0.98 | 0.99 | 1.01 | SI2: Fig. A2.2 |
| SEA04 model 2, vascular plants and bryophytes | Mesh + square | 0.98 | 0.99 | 1.01 | SI2: Fig. A2.3 |
| MEA10, vascular plants only | Square only | 0.97 | 0.98 | 1.00 | SI3: Fig. A3.1 |
| MEA10, vascular plants only | Mesh only | 0.98 | 0.99 | 1.01 | SI3: Fig. A3.2 |

random effects gave a DIC value of 1732.7 compared to a value of 1712.7 for the full model; the preferred model in this case was that with the spatial random field only (DIC = 1709.0). Estimates of the overall random effects variance explained by the structured spatial field varied between 87.4% (MEA10, vascular plants only) to 99.7% (SEA04 Model 1, vascular plant and bryophyte richness combined) across the models including both random effects. All estimated spatial random fields can be viewed in Supplementary Information 1.

The addition of an extra kilogram of total N deposition on plant species richness per hectare per year was consistently amongst the smallest effect estimated by our models, with low uncertainty in comparison to other covariates (Figs. 1A–1C; Supplementary Information 2: Figs. A2.1, A2.2, A2.3). Exponentiating the fixed effects indicated that the impact of increasing total N deposition is equivocal across these datasets and models (Table 2): six models estimated a drop of around 1% in species richness as the most likely impact (Figs. 1B–1C; Supplementary Information 2: Figs. A2.2, A2.3; Supplementary Information 3: Figs. A3.1, A3.2), although two of these models (Supplementary Information 3) were formulated *post hoc* and were not favoured as the best models for the dataset by their DIC estimates. Two models, estimated from MEA10 using the DIC-favoured model, suggested that an increase in species richness had more support (Fig. 1A; Supplementary Information 2: Fig. A2.1; Table 2). The reduced covariate set used for the SEA04 data (model 1) for closer comparison to the covariate set used in the MEA10 analysis made no difference to the estimated effect of N deposition calculated from the larger set of covariates from SEA04 (model 2; Figs. 1B–1C; Supplementary Information 2: Figs. A2.2, A2.3; Table 2). Figure 2 illustrates the estimated partial effect of N deposition (with other covariates set to zero) on vascular plant species richness for two of our models (MEA10 and SEA04 model 1); these demonstrate both the relatively small predicted mean impact on species richness, the disagreement in impact direction between the datasets, and the uncertainty attached to these predictions.

The results presented in SI3 were included as exploratory, *post hoc*, analyses after it was found that our DIC-favoured models of the MEA10 dataset indicated that the most likely

Pescott and Jitlal (2020), *PeerJ*, DOI 10.7717/peerj.9070

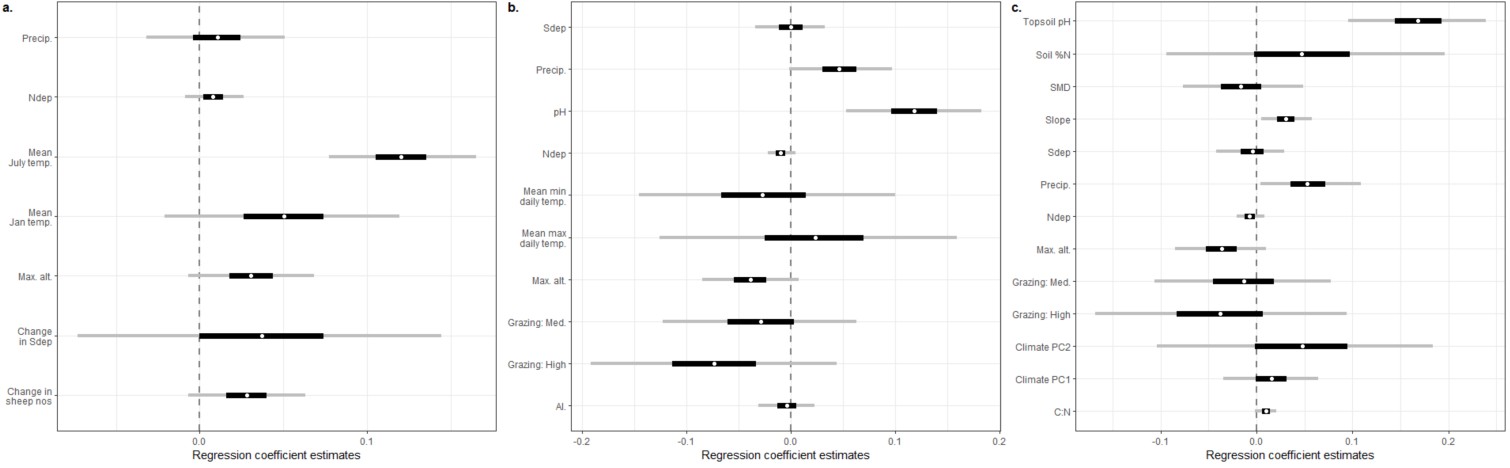

**Figure 1** **Regression coefficient plots.** (A) Estimated regression coefficients for the reanalysis of *Maskell et al. (2010)*; (B) estimated regression coefficients for the reanalysis of *Stevens et al. (2004)* using a reduced set of covariates chosen for their similar ecological status to the covariates used by *Maskell et al. (2010)*, referred to in this paper as SEA04 model 1; (C) estimated regression coefficients for the reanalysis of *Stevens et al. (2004)* using a set of covariates designed to match the original analysis of that paper as closely as possible, referred to in this paper as SEA04 model 2. The dependent variable was vascular plant species richness in all cases. White circles represent the posterior median estimate, black bars the posterior 50% credible interval, grey bars the posterior 95% credible interval. All covariates are described in Table 1.

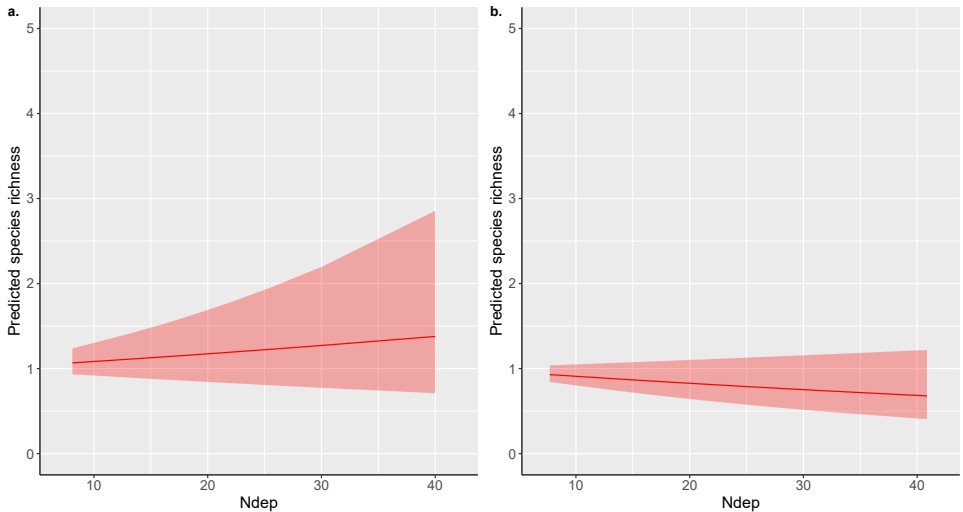

**Figure 2** **Total nitrogen deposition partial effect predictions.** INLA model predictions of the partial effect of total nitrogen deposition (Ndep; kg ha$^{-1}$ yr$^{-1}$) on acid grassland species richness estimated for (A) MEA10 and (B) SEA04 model 1; both the models used for these predictions used vascular plant richness only as the dependent variable. Predictions were estimated for each 10th-percentile of the respective Ndep ranges covered by each study using the linear combinations option of INLA. Linear combinations were estimated for each point in the Ndep range with other covariates set to zero, hence the low values of the predicted richnesses for different levels of total Ndep.

overall influence of Ndep on species richness was more likely to be positive, contrary to the conclusions of *Maskell et al. (2010)*. Our additional analyses (SI3), each with only one of the spatial random effects used in the full model, did show a reversal of average sign, with the analysis matching that of *Maskell et al. (2010)*, i.e., using a 1 km square random effect, indicating the strongest negative effect of N deposition on vascular plant richness (a mean of a 2% loss of species richness per kg ha$^{-1}$ yr$^{-1}$ of total N deposition; Table 2; Supplementary Information 3: Fig. A3.1). However, this model had a DIC value 7.4 units higher than the favoured model.

# DISCUSSION

Previous work on the estimation of the effects of nitrogen deposition on plant species richness from space-for-time substitution studies is likely to have had a significant impact on both science and policy (assuming that the number of citations received can be used as an index to this). Therefore being clear about the reliability and accuracy of estimates from observational work is important, not only for fundamental reasons of improving ecological understanding and analysis, but also because policy decisions must be made concerning the utility of funding work in this area relative to other ways of assessing environmental change (although field experimental work may have issues of its own; *Peters, 1991*; *Phoenix et al., 2012*). Clear and accurate estimates of the likely sizes of effects are invaluable for informing future studies, for example, through power analyses, or the use of informative priors for
Bayesian approaches. Our results suggest that only small effects of nitrogen deposition on species richness may be detectable in these observational datasets.

The headline result of *Stevens et al. (2004)* reported "a reduction of one species per 4-m$^2$ quadrat for every 2.5 kg N ha$^{-1}$ yr$^{-1}$ of chronic nitrogen deposition". An estimated reduction in species richness of 23% (based on a change from deposition of 5 kg N ha$^{-1}$ yr$^{-1}$ to 17 kg N ha$^{-1}$ yr$^{-1}$) was also highlighted as a key message using a regression coefficient estimated from a model including only Ndep as a covariate. The estimate from our SEA04 model 1 implied a loss of around 1% of richness per kg N ha$^{-1}$ yr$^{-1}$; however, this ignores the uncertainty associated with this particular coefficient estimate, which included an estimate of no effect at the 97.5% percentile, and the larger uncertainty associated with alternative model specifications (including ones that we have not explored here). Indeed, our estimates from the larger, random-stratified national survey of *Maskell et al. (2010)* suggested that a small average increase in richness along the total nitrogen deposition gradient was more plausible under the favoured model for that dataset. Furthermore, the partial effects calculated from our full models (Fig. 2) indicated that the average effect on richness across Britain may only approach a difference of around one species in either direction. This is also notable because previous commentaries (e.g., *Tipping et al., 2013*) have suggested that differences in the apparent effect sizes of Ndep between the two observational studies reanalysed here were likely due to study design, whereas our work suggests that the differences are actually minimal, and that those found previously may simply be artefacts of the different statistical modelling procedures adopted by the original studies. The different median signs found in our analyses could also be due to differences in the statistical populations targeted, namely the differential sampling of grassland types, with the greater plant community coverage of the dataset of *Maskell et al. (2010)* potentially covering more sites on the left of a net primary productivity/diversity unimodal curve, resulting in an average increase in richness under fertilisation (see below for more discussion of the potential impacts of non-linear responses and other issues on the current work).

Our results should cause others to re-evaluate their approaches to observational data. For example, *Field et al. (2014)* also used PSVS, and other *P*-value based selection techniques, to formulate models in their space-for-time analyses of Ndep impacts for a number of semi-natural habitats across Britain, as well as highlighting simple univariate relationships through scatter plots. These authors, however, did note in their methods section that their results should be "interpreted with caution", although this statement of uncertainty was not clearly carried through to other parts of their paper, nor to other research. For example, *Payne et al. (2017)* used data from both SEA04 and *Field et al. (2014)* to forecast the impacts of Ndep on plant species richness under future nitrogen deposition scenarios. Ndep and its polynomial transformations were the only covariates in these models. *Payne et al. (2017)* noted in their online supplementary material (their Web Panel S1) that such forecasting makes the strong assumption that covariances between variables remain the same at the future time point (i.e., the distribution of nitrogen deposition amounts will vary with, e.g., mean maximum July temperature in the same way in 2030 as it did when originally modelled). However, they did not clearly acknowledge that the Ndep-only models in

their forecasting exercise had received no validation in support of their status as the best predictive (or explanatory, for that matter) models by the original studies or elsewhere. Ndep may be a useful covariate for producing good predictive models of species richness in certain habitats, but this should be demonstrated using measures of out-of-sample predictive ability, and the resulting partial effects of Ndep in such models have no intrinsic claim to be reliable estimates of its causal relationship with richness (cf. *Payne et al., 2017*).

These issues from the literature should not detract from the fact that there are also several ways in which the work presented here could potentially be improved upon. Our efforts are linear models (as for SEA04, MEA10, *Field et al., 2014*, among others), but there is evidence that the response of species richness to Ndep may be better modelled as unimodal (*Tipping et al., 2013*; *Simkin et al., 2016*; *Clark et al., 2019*). *Simkin et al. (2016)* still, however, reported declines in the plant species richness of open habitats above 8.7 kg N ha$^{-1}$ yr$^{-1}$, very similar to the 7.9 kg N ha$^{-1}$ yr$^{-1}$ richness decline threshold identified for acid grassland by *Tipping et al. (2013)*, suggesting that a large part of the gradient studied by SEA04 and MEA10 may still be well-approximated by a linear relationship (although we note that the approach of *Tipping et al. (2013)* also makes the assumption that there are no omitted linear or non-linear variables correlated with Ndep that might change their estimated univariate breakpoint relationship). Indeed, the linear component of a non-linear trend is often considered the most policy-relevant summary by those who routinely produce ecological indicators (e.g., *Soldaat et al., 2017*). Different approaches to accounting for this likely non-linearity in multivariable models, such as the inclusion of interactions or smoothers, could however be further explored, particularly given that novel methods focused on causal inference that can account for these issues continue to be developed (*Dorie et al., 2019*).

Other uncertainties relating to our conclusions pertain to the fact that the broad-scale spatial field used here may be accounting for information that, if known, would change the size of the Ndep impact regression coefficient. This could be in the form of additional covariates, or more highly resolved estimates of the Ndep load that a location has actually received. The Ndep estimates used here (and by the original studies) are resolved to a grain size of 5 × 5 km, and this additional uncertainty could have attenuated our estimate of the regression coefficient (the measurement error in explanatory variables problem; e.g., see *Fox, 2016*), even if there is no systematic bias relating to the association of 5 × 5 km-estimated Ndep levels with particular types of vegetation. This argument is likely to apply to many of the variables used here and in other studies, particularly given that many are estimated from other modelling exercises at relatively large spatial grain sizes (Table 1). Conversely, measurement error coupled with conditioning on statistical significance, as happens through PSVS (e.g., *Stevens et al., 2004*; *Field et al., 2014*), is likely to result in the overestimation of effects (*Gelman & Carlin, 2014*; *Loken & Gelman, 2017*). Whether or not the potential for these biases is more serious for inference than the absence of covariates that are unavailable, such as historic land management events (e.g., *Rackham, 1986*), is difficult to say.

The new observational results presented here are in line with much of the experimental literature on Ndep impacts. For example, *Phoenix et al. (2012)* reviewed a group of nine

experiments conducted across the UK, with N treatments which had been running for lengths of time between 7 and 22 years at the time of review. These studies aimed to examine the impacts of "modest treatment doses and avoid[ed] single dose or solid form applications" (*Phoenix et al., 2012*) in order to overcome previous criticisms relating to (potentially unrealistic) very high experimental loadings. These nine studies reported no effects of their experimental N treatments on higher plant richness (although these conclusions of no effect were all based on thresholding *P*-values); two sites indicated increases in richness using the calculated accumulated Ndep dose over the duration of an experiment, although these cases were discounted by *Phoenix et al. (2012)* as being of either minor ecological significance or transitory. *Phoenix et al. (2012)* put forward several reasons why the results from this series of experiments may not reflect the true impact of chronic Ndep on plant richness: for example, sensitive species might already have been lost prior to the establishment of an experiment, and/or an experiment might not have been running long enough for the impacts to have been fully realised. Of course, important changes in community structure can also occur without species loss (*Hillebrand et al., 2018*).

Longer running studies, such as the Park Grass experiment at Rothamsted, England (1856 to the present day; *Silvertown et al., 2006*), avoid some of these criticisms. Although the Park Grass plots that have received experimental N addition have received very high doses, with annual N fertiliser doses starting at 48 kg N ha$^{-1}$ yr$^{-1}$ (*Storkey et al., 2015*), control plots having received only atmospheric deposition may be a useful comparator for some habitats in the wider landscape. Plot 3 of the Park Grass experiment, for example, has in theory only ever received ambient Ndep (*Storkey et al., 2015*); *Lawes & Gilbert (1880)* reported an average of around 48.5 plant species (including bryophytes) in the 0.5 acre ($\sim$2,000 m$^2$) neutral grassland plot between 1862 and 1877; the average between 1939 and 1948 was 34 (*Brenchley & Warington, 1958*), indicating a decline over this period that preceeded the local increase in Ndep (see the first figure of *Storkey et al., 2015* for the local Ndep trend at this location over the 20th century). Unfortunately, the local increase in Ndep at Rothamsted coincided with the decision to split the plots into different liming treatments (*Williams, 1974*; *Storkey et al., 2015*) meaning that subsequent richness estimates cannot be unambiguously compared to the earlier numbers (there is confounding of the Ndep increase with the application of lime, and the area that could be directly compared shrinks to a quarter of the original plot, introducing the need for species–area adjustments). Over the period 1991–2012, however, survey data indicate that there may have been a very slight recovery in richness in Plot 3 (although Storkey et al. present no statistics for this trend), coinciding with a reduction in the N composition of the plot herbage and increases in Simpson's diversity index (*Storkey et al., 2015*).

Overall, then, experimental data using realistic applications of Ndep appear to support our finding that richness is a relatively insensitive metric of such impacts (see also *Hillebrand et al., 2018*). Finally, and to avoid any misunderstanding as to the thesis being presented in this paper, we note that we are *not* stating that overall eutrophication from all sources (e.g., livestock, local fertiliser drift etc.) is unimportant for the conservation of plant biodiversity. General signals of eutrophication in plant communities are widespread and beyond doubt

(e.g., *Smart et al., 2003b*; *Willi, Mountford & Sparks, 2005*). We are forced to conclude, however, that the contribution of Ndep to this phenomenon appears to be smaller, and more uncertain, than many previous analyses of space-for-time Ndep gradients have concluded (cf. *Stevens et al., 2004*; *Field et al., 2014*; *Payne et al., 2017*). It is possible that this is due to the fact that earlier losses of species due to accumulated historic deposition means that the remaining opportunity to detect effects in recent space-for-time studies is limited; this suggests, however, that richness should not be used as an indicator of Ndep impacts, and previous work taking this approach (particularly where highly significant statistical concerns abound) should no longer be cited in support of general conclusions regarding these impacts.

## CONCLUSIONS

The evidence for large negative impacts of nitrogen deposition on plant species richness put forward through analyses of observational data appears to have been overstated. We estimate a possible decline in richness of around 1% per kg ha$^{-1}$ yr$^{-1}$ of total N deposition from two spatially and temporally separated British space-for-time gradient studies, considerably less than the estimates implied previously by described and/or plotted relationships from primary studies (*Stevens et al., 2004*; *Maskell et al., 2010*). Moreover, even this estimate appears uncertain, and our favoured model for the acid grassland data of *Maskell et al. (2010)* suggests that an average increase in richness of a similar magnitude may be more likely. The previous lack of presented regression coefficients derived with causal inference as their main focus, and of models that account for broad-scale spatial autocorrelation, is important to note, because scientists wishing to use the estimated effects of Ndep for the design of future studies, or for the construction of informative priors in new analyses, may be misled as to the size of effect that is expected to be detectable in datasets of this type. The various models presented here could be thought of as a small section of the "multiverse" of potential approaches to these data, an approach that has been put forward as an additional route to transparency and reproducibility in science, and which can provide insights into the fragility or robustness of particular conclusions (*Steegen et al., 2016*). As such this work is unlikely to be the last word on these datasets, or in the general area of observational studies of nitrogen deposition impacts. We hope that our re-analyses inspire further efforts to accurately extract the maximum available knowledge from these valuable datasets, whether for explanatory or predictive purposes, and that evidence synthesis in this area takes these uncertainties and methodological issues into greater account going forwards.

## ACKNOWLEDGEMENTS

We thank the authors of the original studies reanalysed for publishing their data or otherwise making it available to us. We also thank A Britton, M Grainger, P Henrys, SM Smart, LC Maskell for useful comments on the manuscript.

### Funding

The initial work that led to the current manuscript was funded by the Joint Nature Conservation Committee to the UK Centre for Ecology & Hydrology under project NEC06730 (National Plant Monitoring Scheme). This work was also supported by the Natural Environment Research Council award number NE/R016429/1 as part of the UK Status, Change and Projections of the Environment (UK-SCAPE) programme delivering National Capability. The funders had no role in study design, data collection and analysis, decision to publish, or preparation of the manuscript.

### Grant Disclosures

The following grant information was disclosed by the authors:
Joint Nature Conservation Committee to the UK Centre for Ecology & Hydrology: NEC06730.
Natural Environment Research Council: NE/R016429/1.
Change and Projections of the Environment (UK-SCAPE) programme delivering National Capability.

### Competing Interests

Oliver L. Pescott is an employee of the UK Centre for Ecology & Hydrology. Mark Jitlal was an employee of the Wolfson Institute for Preventative Medicine at the time of first submission. Neither has any other competing interests to declare.

### Author Contributions

- Oliver L. Pescott conceived and designed the experiments, analyzed the data, prepared figures and/or tables, authored or reviewed drafts of the paper, and approved the final draft.
- Mark Jitlal conceived and designed the experiments, authored or reviewed drafts of the paper, and approved the final draft.

### Data Availability

RStudio project for the reanalysis of the *Stevens et al. (2004)* dataset is available as a Supplemental File. The reanalysis of the UK Countryside Survey data is not shared, because the authors do not have permission to share the survey locations, which are an intrinsic part of the reanalysis. See the link to the UK Countryside Survey data access policy below for more information.

The data of *Stevens et al. (2004)* are available in *Stevens et al. (2011a)* (DOI: 10.1890/11-0115.1), and also within the RStudio project shared here.

UK Countryside Survey data access policies can be read at https://countrysidesurvey.org.uk/content/data-access.

## Supplemental Information

Supplemental information for this article can be found online at http://dx.doi.org/10.7717/peerj.9070#supplemental-information.

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
