# Peer review of "Reassessing the observational evidence for nitrogen deposition impacts in acid grassland: spatial Bayesian linear models indicate small and ambiguous effects on species richness"

_PeerJ, doi:10.7717/peerj.9070_

## Round 0.1 · original submission · Minor Revisions

I agree with the reviewers that this is a good paper, well structured and supported.

Reviewers ask for some minor changes, including It would the possibility of including the R code so that the analysis can be directly replicated.
Also, one of the referees would ask for additional discussion about the spatial structures in the datasets and consequent implications in the outcomes.

·

Basic reporting

This is a well presented and scholarly article describing the results of a re-analysis, using new statistical techniques, of existing datasets describing acid grassland species richness, with a view to detecting and attributing the impacts of nitrogen deposition. I found the English clear and the article well written and easy to follow, especially given the technical statistical nature of content. The article structure was logical, and figures appropriately selected and clearly labelled.
I had only a few minor comments on the presentation of the text:
1. Reference citations in the text don’t seem to be consistent about the number of authors before ‘et al’ or the way the names are presented (some have both first and family names) this should be checked throughout the document.
2. Ln 133. The number 320 should refer to plots not sites.
3. Figures 1, 2 & 3 – It’s instructive to compare these different models to see how they change between datasets and with the inclusion of different parameters – this would be easier if the 3 panels could be presented side by side on a landscape format page.
4. Figure 4 – what are the units of species richness change?

Experimental design

The study describes the application of a new statistical approach to existing datasets with the aim of comparing the outcome of different approaches to the analysis of these data. The new approach aims to be more rigorous and transparent in the model building process. The rationale behind this re-analysis is clearly explained and a rigorous investigation has been carried out to explore the effects of N deposition using this new approach. The statistical methodology is well described, and appears appropriate, although please note that I am not an expert in Bayesian statistics and so I am unable to comment in detail on this aspect.
One thing I did note is that in the analysis of the SEA04 dataset the authors made the decision to combine four of the climatic variables, which were closely correlated, by using the first two components of a PCA of these data instead. It would be useful to see a fuller description of the correlation between all of the variables used in the models, since e.g. total N and total S deposition are likewise usually closely correlated, was there a cut-off used to decide how correlated variables could be before they were no longer considered independently?

Validity of the findings

The conclusions drawn from the work appear valid and are clearly linked to the results. One thing that struck me was that the spatial random fields had quite different structures between the two datasets. You mention in the discussion that these fields may include effects related to the limited resolution of some of the co-variables (e.g. N deposition data are represented only at the 5km grain size) and other processes such as dispersal. But, given the general similarities of the vegetation composition and explanatory variable datasets, why would the spatial structure in the data be so different? It may be worth expanding on this slightly in the discussion.
I also think it would be worth adding some discussion on how these results compare to those derived from experimental studies (since most other analyses of observational datasets are subject to the criticisms being made by the present manuscript). Do the small effects of N on richness match responses to experimentally applied N? I was particularly curious about this given the large effect of soil %N SEA04 model 2.

Additional comments

In general, I think that this is a well written paper, which makes a useful contribution to the debate around attribution of N impacts on biodiversity. The observations made in the paper are widely applicable across a range of studies and will make a useful and valuable contribution to the literature.

·

Basic reporting

No comment

Experimental design

My only comment would be about imputed mean values (Line 164 in the Methods section). Did you do any sensitivity analysis of using other imputation approaches - are these plots likely to be close to the mean value? Is there an option for multiple imputation approach (perhaps to look at the sensitivity of the model to these imputed values)? This is a minor comment.

Validity of the findings

It would be good to have your R code so that the analysis can be directly replicated. I assume this will be available to the readership online in some form (OSF, Github, etc)

Additional comments

Wow! It was a pleasure to read this manuscript. Thank you for the opportunity. Excellent, well written with appropriate referencing, style and structure. The logical and detailed approach to reassessing the published data was very impressive. The world needs more papers like this! I do have a few minor comments that I think might improve the manuscript (I would wouldn't I!) Throughout you seem unable to believe your own findings! For example 235:242, you wait until the last line to point out the lack of support for this model that you describe in quite some detail. In line 296:324 you suggest how your approach is "deficient" (potentially). I am supportive of caution, but I think you could be more positive about your findings. I find it hard (I am not knowledgable about ndep) to understand 326 where you say that you do not dispute the negative impacts of nitrogen but you show that there is little evidence of such an effect. I think I understand this but it would help (me at least) to have an understanding of what is a "heavy" nitrogen deposition. Is this something we are likely to see in Agricultural settings in the UK? Or is it only possible in controlled (lab) experiments. Related to this is that there is a need (again for me) to have a paragraph on the consequences of your finding for policy or Research priorities. If our new Government asks you what to do about the problem of ndep in the UK what will you advise them?

---

## Round 0.2 · accepted · Accept

Both reviewers are very satisfied with the job made by the authors. So, even though a couple of very minor changes are still requested I think that the paper can be accepted for publication.

·

Basic reporting

I had only a couple of small comments on the revised text:
1. Line 100 – presumably pH, Al and C:N refer to soil parameters? This needs to be specified as it is the first mention of them.
2. Line 106 – after N dep it should be ‘than’ not ‘that’.
3. Line 109 – hyphenate pre-treatment.

Experimental design

No comment

Validity of the findings

No comment

Additional comments

The authors have dealt satisfactorily with all of my comments on the previous version of their manuscript. Their additions to the discussion particularly have much improved the paper by placing it into better context of both the experimental and observational evidence for N deposition impacts.

·

Basic reporting

No additional comments

Experimental design

No additional comments

Validity of the findings

No additional comments

Additional comments

Great paper - no additional comments. The authors have done a great job and answered all my queries